# Functional Dental Pulp Regeneration: Basic Research and Clinical Translation

**DOI:** 10.3390/ijms22168991

**Published:** 2021-08-20

**Authors:** Zhuo Xie, Zongshan Shen, Peimeng Zhan, Jiayu Yang, Qiting Huang, Shuheng Huang, Lingling Chen, Zhengmei Lin

**Affiliations:** Department of Operative Dentistry and Endodontics, Guanghua Hospital of Stomatology, Guangdong Provincial Key Laboratory of Stomatology, Sun Yat-sen University, Guangzhou 510055, China; xiezh23@mail2.sysu.edu.cn (Z.X.); shenzsh@mail2.sysu.edu.cn (Z.S.); zhanpm@mail2.sysu.edu.cn (P.Z.); yangjy53@mail2.sysu.edu.cn (J.Y.); huangqt25@mail.sysu.edu.cn (Q.H.); hshuh@mail2.sysu.edu.cn (S.H.)

**Keywords:** dental pulp stem cells (DPSCs), dental pulp regeneration, tissue engineering, clinical translation, regenerative endodontics

## Abstract

Pulpal and periapical diseases account for a large proportion of dental visits, the current treatments for which are root canal therapy (RCT) and pulp revascularisation. Despite the clinical signs of full recovery and histological reconstruction, true regeneration of pulp tissues is still far from being achieved. The goal of regenerative endodontics is to promote normal pulp function recovery in inflamed or necrotic teeth that would result in true regeneration of the pulpodentinal complex. Recently, rapid progress has been made related to tissue engineering-mediated pulp regeneration, which combines stem cells, biomaterials, and growth factors. Since the successful isolation and characterisation of dental pulp stem cells (DPSCs) and other applicable dental mesenchymal stem cells, basic research and preclinical exploration of stem cell-mediated functional pulp regeneration via cell transplantation and cell homing have received considerably more attention. Some of this effort has translated into clinical therapeutic applications, bringing a ground-breaking revolution and a new perspective to the endodontic field. In this article, we retrospectively examined the current treatment status and clinical goals of pulpal and periapical diseases and scrutinized biological studies of functional pulp regeneration with a focus on DPSCs, biomaterials, and growth factors. Then, we reviewed preclinical experiments based on various animal models and research strategies. Finally, we summarised the current challenges encountered in preclinical or clinical regenerative applications and suggested promising solutions to address these challenges to guide tissue engineering-mediated clinical translation in the future.

## 1. Introduction

Pulpal and periapical diseases, which are two of the most prevalent oral diseases, typically result from irreversible damage to the dental pulp as a result of intense and severe stimuli, such as dental caries, accidental trauma, or iatrogenic causes [1,2]. Dental pulp is a highly vascularised and innervated tissue located within rigid dentinal walls, and it performs several functions, such as responding to external signals, providing nutrition and ameliorating neuronal sensitivity by repairing pulp through mineralisation [3]. Therefore, loss of this tissue results in loss of tooth vitality and requires endodontic treatment. Root canal therapy (RCT) is a classical and effective treatment that is currently utilised in dental practice, offering high success rates for pulp and periapical diseases; however, teeth after RCT are susceptible to altered pulp defense and sensory function, even fractures, as a consequence of pulp loss [4]. Although apexification is used to be a conventional treatment in which either calcium hydroxide paste or mineral trioxide aggregate (MTA) is utilised to generate apical barriers in the immature teeth after pulp necrosis, it may result in susceptibility to root fractures due to thin canal walls and poor root/crown ratio, as well as long-term calcium hydroxide placement which may weaken the dentin and induce root fracture [5].

Murray et al. [6] proposed the term “Regenerative endodontic treatment (RET)” in 2007, based on a tissue engineering concept. The 2016 American Association of Endodontists (AAE) guidelines formally defined RET as a collection of “biologically based procedures designed to replace damaged tooth structures, including dentine and root structures, as well as cells of the pulp-dentine complex” [7]. The 2018 AAE guidelines pointed out that pulp revascularisation was currently the only RET officially used in the clinic [8]. Formally approved and used in the clinic, pulp revascularisation may be defined as the revascularisation of an immature permanent tooth with an infected necrotic pulp and apical periodontitis/abscess. It can promote the root development and reinforcement of dentinal walls by deposition of hard tissue, providing an additional treatment option [9,10]. Nevertheless, pulp revascularisation is not synonymous with pulp regeneration because the regenerative tissue is not similar to that of native pulp [11]. Pulp regeneration implies restoration of the lost or damaged part of the original dental pulp tissue, leading to the complete reestablishment of biological function. According to AAE 2016, the degree of success of functional pulp regeneration is largely measured by the extent to which it is possible to attain primary, secondary, and tertiary goals: (1) primary goal: the elimination of symptoms and evidence of bony healing, (2) secondary goal: increased root wall thickness and/or increased root length, and (3) tertiary goal: positive response to vitality testing [7]. In recent years, the exploration of effective therapeutic strategies has attracted attention to achieve the goals of this regenerative procedure.

Exciting new insights into the fundamental underpinnings of pulp regeneration are now being made in an era in which tissue engineering pipelines offer the potential to specifically target strategies of significance [12]. In general, tissue engineering involves stem cells, biomaterials, and bioactive growth factors. Stem cells are responsible for tissue healing and regeneration after damage [13]. Biomaterials can not only provide three-dimensional growth space for cells but can also regulate cell function in the local niche [14]. Growth factors enhance the regenerative effect and regulatory function of stem cells [15]. The effect of each element on pulp regeneration is different, but all are considered important for the final result. Successful pulp regeneration requires manipulation of a reparative microenvironment via a suitable combination of these three components [12,16].

Currently, a variety of studies are underway to evaluate tissue engineering-based strategies for pulp regeneration. Here, we summarise the latest advances in functional pulp regeneration mainly from the perspective of applicable cell sources, natural/synthetic biomaterials, growth factors, and different regenerative approaches based on multiple combinations of these elements. Furthermore, we elaborate on a series of preclinical explorations via various animal models and research patterns and then propose several optimised strategies from the standpoint of efficiency and safety. Potential solutions and promising ideas to be tested that can address challenges and obstacles in clinical application are also discussed. We hope that our review can provide a reference with which to direct dental stem cell-mediated clinical transformation in the future.

## 2. Literature Search and Scope of the Review

An electronic search was performed on PubMed, Web of Science, and Scopus Library databases of English language publications from inception to 20 April 2021. The related key words included “Dental pulp stem cells (DPSCs)”, “Dental pulp regeneration”, “Dental pulp repair”, “Tissue engineering”, “Clinical translation”, and “Regenerative endodontics”. However, due to the scope and extent of this search, a wide-ranging comprehensive narrative review of functional dental pulp regeneration rather than a systematic review was undertaken.

## 3. Treatment Status and Development Trend of Pulpal and Periapical Diseases

### 3.1. RCT

Population-based studies world-wide indicated that approximately 48% of healthy individuals have endodontic lesions [2]. RCT, appearing at the beginning of the 20th century, aims to minimise the space for microbial reinfections and create a beneficial healing environment for periapical tissues by mechanically and chemically disinfecting root canal systems and then sealing the area with inert materials [4]. With the development of related instruments and materials, as well as the improvement of operative techniques, RCT has become the most classic treatment method for pulpal and periapical diseases, comprising a set of complete treatment processes and prognosis evaluation programs. However, it is also important to be aware of the complications and limitations of RCT (Table 1):Technology sensitivity: it is difficult for general dentists to properly prepare and fill the complex and changeable root canal systems [17]; thus, they are likely to cause complications, such as perforation, instrument fracture, underfilling, and overfilling.Reinfection: root canal reinfection caused by root canal sealant dissolution and crown microleakage accounts for approximately 60% of RCT complications [18].Loss of pulp function: along with large hard tissue defects, denervation and avascularity, damaged teeth after RCT lose almost all physiological pulp functions, such as sense, nutrition, and defence, and are susceptible to fractures [19].Age limitations: immature permanent teeth cannot continue to develop after RCT; in addition, calcification and obliteration of root canals in elderly individuals may inevitably increase operative difficulty [20].

Therefore, more effective clinical strategies are warranted to develop treatments for pulpal and periapical diseases.

### 3.2. Pulp Revascularisation

Pulp revascularisation is a promising alternative approach in pulpal and periapical diseases; it is different from RCT with respect to the basic concept and operative procedures. Pulp revascularisation requires sufficient chemical disinfection using irrigants and intracanal medicaments rather than mechanical debridement, with a focus on inducing bleeding into the root canal to provide a favourable regenerative niche [5]. Lin et al. [23] reported that pulp revascularisation in immature permanent teeth with apical periodontitis showed a better outcome than calcium hydroxide apexification with respect to increased root thickness (81.16% in pulp revascularisation group; 26.47% in apexification group) and root length (82.6% in pulp revascularisation group; 0% in apexification group), but pulp revascularisation had no obvious advantage in inducing closure of the apical foramen. A prospective analysis of 16 consecutive cases was conducted to assess the apical closure of pulp revascularisation, and the results indicated that closure was incomplete in 47.2% of cases and complete in 19.4% of cases [24]. Much effort has been made to improve clinical efficacy, such as adding platelet-rich plasma (PRP), platelet-derived growth factor (PDGF) and other active factors [25].

Undoubtedly, pulp revascularisation has significantly improved treatment strategies, providing a minimally invasive approach and infection control; however, it still has some complications and limitations (Table 1) [21,22,23]. Although it has been used clinically, pulp revascularisation cannot be referred to as pulp regeneration. New tissue formation in the canal space after pulp revascularisation is characterised as bone-, cementum-, and periodontal ligament-like tissue, not pulp/dentin-like tissue [11].

### 3.3. Clinical Goals of Pulp Regeneration

Dental pulp is a highly ordered connective tissue that is divided into the dentinoblastic zone, cell-poor zone, cell-rich zone, and central zone from outside to inside. The dentinoblastic zone forms a functional pulpodentinal complex, which is the frontier of reparative dentin formation and exhibits a protective response to exterior stimuli. Rich blood vessels and nerve fibres provide nutrition and sensory function for the pulp.

Therefore, ideal pulp regeneration refers to regenerating natural and precise structures, including (1) the pulpodentinal complex, (2) rich blood vessels and nerve fibres, and (3) resorbed root or cervical or apical dentin, eventually achieving pulpodentinal complex reconstruction with angiogenesis and neurogenesis, as well as physiological function rehabilitation, namely, nutrition, sensation, and immunological defence (Figure 1).

## 4. Biological Studies of Pulp Regeneration

Tissue engineering techniques generally employ various combinations of the interplay of stem cells, biomaterials, and growth factors to promote functional restoration of tissue structure and physiology in impaired or damaged tissues [12]. A suitable combination of these three elements enables the manipulation of a biomimetic microenvironment containing a well-functioning vascular system, which ensures nutrient supply, waste removal, inflammatory response, and subsequent regeneration for the pulp (Figure 2).

Dental stem cells can be used as effector cells because of their multidirectional differentiation potential, and they can secrete exosomes and cytokines to regulate other niche cells through paracrine effects.
(1)Biomaterials can not only provide a three-dimensional growth space for cells but can also regulate the paracrine secretion of stem cells and the phenotypic switching of macrophages in the local niche.(2)Growth factors enhance the regenerative effect and regulatory function of stem cells and niche cells.(3)We can choose different combinations of these elements that are applicable to different situations to achieve the best therapeutic effect (Table 2).

### 4.1. Stem Cells Applicable for Pulp Regeneration

Stem cells are integral parts of modern tissue engineering aimed at regeneration, especially when the damage is too extensive to self-regenerate or taking cell-free approaches. Cell based-pulp regeneration is similar to the general understanding of tissue engineering concepts that adopts cell-based therapies, and requires a cell source delivered into the host for tissue to regenerate to its original or close to its original state [22]. Various application strategies of stem cells are promising to regenerate a pulpodentinal complex and to promote the translation of laboratory experiments into clinical applications.

#### 4.1.1. DPSCs and Other Stem Cell Sources

Suitable stem cells are the key element for pulp regeneration. Shi et al. isolated and identified stem cells from dental pulp tissues of permanent and deciduous teeth, called DPSCs and SHEDs, respectively; soon afterward, other mesenchymal stem cells derived from dental tissues, such as periodontal ligament stem cells (PDLSCs), dental follicle precursor cells (DFPCs), stem cells from the apical papilla (SCAPs), and Hertwig’s epithelial root sheath (HERS), became known as a broader population [35,36,37].

DPSCs have strong self-renewal ability and multidirectional differentiation potential, and it is easier to induce their differentiation into odontoblasts than to induce the differentiation of other dental stem cells due to tissue specificity. Furthermore, DPSCs can differentiate into endothelial and nerve cells and secrete many regulatory proteins for angiogenesis and neurogenesis [38,39]. Compared with PDLSCs and SCAPs, DPSC sheets can form loose connective tissue similar to native pulp tissue [40]. All these studies suggest that DPSCs are an ideal cell source for pulp regeneration (Table 3).

Stem cells from bone marrow (BMSCs) and adipose tissue (ADSCs) have pulp regenerative potential as well [43,44]. Even though their regenerative capacity for pulp tissues, angiogenesis, and reinnervation was weaker than that of DPSCs, BMSCs and ADSCs may be alternative cell sources when the supply of autologous pulp tissues is limited due to some pathological conditions, such as inflammation and senescence [43]. A list of the possible advantages and disadvantages of these applicable stem cells is presented in Table 4.

#### 4.1.2. Surface Markers, Subpopulations and Side Populations of DPSCs

Ninety-five percent of human MSCs express the surface antigens CD105, CD73, and CD90/Thy-1, while the most commonly reported negative markers are CD11b, CD14, CD19, CD34, CD45, CD79a, and HLA-DR [57]. Similar to the case for MSCs, there are no specific surface markers by which to exclusively identify DPSCs, which belong to a versatile and heterogeneous population containing multiple subpopulations that differ in biological activities and functions. Thus, it is necessary to isolate these different subpopulations by specific surface markers to provide suitable cell resources for clinical use [58] (Table 5).

Several surface markers and related subpopulations have been identified from heterogeneous DPSCs in succession. It was found that 30% of DPSCs expressing vascular endothelial cell growth factor receptor-2 (flk-1/VEGF-R2) also expressed other vasculogenic markers, and flk-1+ DPSCs could contribute to the regeneration of vascularised pulp tissues through synergistic differentiation into osteoblasts and endothelial cells [59]. CD31-/CD146- and CD105+ cells exhibit increased expression of angiogenic/neurotrophic factors with a higher trophic effect on migration, immunomodulation, antiapoptosis, and angiogenic, neurogenic, and regenerative potentials compared with that of total DPSCs [13,58,60,61]. DPSCs expressing certain biomarkers, such as STRO-1, CD271, CD90, αSMA, and programmed cell death-1 (PD-1), exhibit longer viability and better multipotentiality [62,64,65,66,67,68]. Accordingly, the quality and quantity of DPSCs used for clinical applications can be improved by cell sorting and purification strategies.

The supply of pulp tissues is extremely limited in aged dental pulp, and DPSCs from aged pulp show decreased proliferative and differential capacities. Thus, it is urgent to isolate and identify targeted regenerative subpopulations for the ageing population. Chen et al. identified a single group of CD24a+ cells from mouse dental papilla called multipotent dental pulp regenerative stem cells (MDPSCs) that exhibited enhanced osteogenic/odontogenic differential capabilities and could regenerate pulpodentinal complex-like tissues in vivo. CD24a+ cells seem to be an excellent alternative cell source for highly efficient pulp regeneration, especially with respect to cell viability in elderly individuals [69].

It is still unclear whether the regenerative potential of MSCs is conferred by only one subpopulation or several subpopulations; however, there is no doubt that the application of specific DPSC subpopulations favours the translational management of standardised clinical products as well as the stability between batches, eventually improving the success rate of pulp regeneration [69].

#### 4.1.3. Molecular Mechanism Underlying the Multipotent Differentiation of DPSCs

DPSCs, with multilineage differentiation potential, can differentiate into various types of cells under transcriptional, posttranscriptional, and epigenetic regulation. Of note, the differentiation program of DPSCs can be regulated by the complicated regulatory networks built with these signalling processes rather than being determined by a single element [70]. For example, Küppel-like factor 4 (KLF4), a critical transcription factor, gained much attention after it was shown in 2006 to be one of four factors required for induced pluripotent stem (iPS) cells. Several lines of evidence exist to support the regulatory networks acting in the odontoblast differentiation of DPSCs (Figure 3):
(1)Transcriptional regulation: KLF4 could directly upregulate the expression levels of odontoblastic-related genes in DPSCs, such as Dmp1, Dspp, and Sp7, by binding to their promoters during odontoblastic differentiation of DPSCs [71,72]. In addition, nuclear factor I-C (NFIC) could bind directly to the Klf4 promoter and stimulate Klf4 transcriptional activity, thereby regulating Dmp1/Dspp signalling during odontoblast differentiation [73]. Similarly, the transcription factor SP1 could regulate KLF4 through a binding site lying in a CpG island in the promoter region of Klf4 [74].(2)Posttranscriptional regulation: Competitive endogenous RNAs (ceRNAs), a group of transcripts, can affect Klf4 mRNA by competitively binding to miRNA response elements. Sp1 functions as a ceRNA of Klf4 during odontoblast differentiation by competing for miR-7a, miR-29b, and miR-135a [75].(3)Epigenetic modification: As mentioned above, transcription factors such as KLF4 and SP1 mainly exert their functions by binding to specific DNA motifs. Epigenetic modifications of these motifs might alter DNA accessibility and thereafter affect the expression of binding transcription factors and downstream genes [74]. For example, the demethylation of the SP1/KLF4 binding motif during odontoblastic differentiation enhanced the efficiency of SP1 binding and transcriptional regulation of KLF4. Additionally, KLF4 could recruit the histone acetylases P300 and HDAC3, which relaxed and provided a more open chromatin structure to transactivate the expression of Dmp1 and Sp7 [76]. Furthermore, a recent study indicated that TET1 can potentially promote odontoblast differentiation by inhibiting FAM20C hydroxymethylation and subsequent transcription [77].

It seems feasible that the upregulation of KLF4 or its promoters could promote mineralisation events during pulpal repair processes. However, activation of excessive transcriptional targets of KLF4 may lead to adverse side effects and ectopic interactions. Future efforts should focus on the precise mechanisms of KLF4 in vivo.

In addition to their odontoblast differentiation capacity, DPSCs have exhibited multidirected differentiation potential into, for example, endothelial cells, neurons, cardiomyocytes, myocytes, and hepatocyte-like cells [78]; therefore, identifying cellular regulators that control stem cell fate is critical to devising novel treatment strategies.

### 4.2. Biomaterials

As carriers of cell transplantation, biomaterials provide a three-dimensional space for DPSC growth and metabolism. Moreover, biomaterials with suitable bioactive cues may create a favourable microenvironment to fine-tune the self-renewal capacity and multipotentiality of DPSCs, thus promoting tissue regeneration. Currently, scaffold materials, both natural and synthetic, are now widely used in pulp regeneration.

#### 4.2.1. Natural and Naturally Derived Substances

Since blood clots induced in pulp revascularisation might be one of the earliest natural substances to develop, naturally derived substances such as collagen, fibrin, gelatine, chitosan, hyaluronic acid, alginate, and peptide-based scaffolds have been investigated as scaffolds for pulp regeneration because they are highly biocompatible, biomimetic, readily available, inexpensive, and easy to fabricate into hydrogels [14,79].

Decellularised extracellular matrix (ECM) serves as a type of biological scaffold with complete elimination of donor cells and antigens while preserving the ECM structure; it is able to provide a suitable regenerative niche for exogenous and endogenous stem cells [80,81]. Human decellularised ECM hydrogel significantly promotes odontoblastic, neurogenic, and angiogenic differentiation of DPSCs [82]. Considering its minimally invasive injectability, in situ gelation, and adjustable degradation, this tissue-specific hydrogel may hold great translational potential for pulp regeneration [83]. Decellularised pulp tissues from humans, bovines, swine, and rats have been applied in dental regeneration [84,85,86]. Species and age can not only impact the performance of an ECM hydrogel but can also have strong implications concerning scalability, a current primary challenge in tissue engineering. Further research should focus on its feasibility, safety, and efficacy in vivo.

The dentin matrix may be another promising natural biomaterial with which to regenerate the pulpodentinal complex due to its nonimmunogenicity and richness in dentinogenesis-related growth factors [87]. Some possible application forms have been proposed, such as nanostructure demineralised dentin matrix (DDM) treated with different demineralised agents, extracted dentin matrix components, and treated dentin matrix paste (TDMP) [88,89,90,91]. TDMP possesses better chemical and biological characteristics than calcium hydroxide (CH) in vitro and in vivo, and is likely to be a novel pulp capping agent [91].

#### 4.2.2. Synthetic Polymer Materials

Several synthetic polymers, such as polylactic acid (PLA), poly(l-lactic) acid (PLLA), polyglycolic acid (PGA), poly(d,l-lactide-coglycolide) (PLGA), and polycaprolactone (PCL), have been suggested as potential scaffolds in clinical dental applications due to their nontoxic, biodegradable and operable properties, including alterable mechanical stiffness and degradation rate [14]. However, bioscaffolds used for pulp regeneration must be easily introduced in narrow spaces and possess perfectly controlled porosity, which becomes a limiting factor for traditional synthetic polymers in clinical applications.

Fortunately, great advances have been made in the recreation of pro-regenerative ECM mimics using synthetic materials. As an injectable scaffold of DPSCs, nanofibrous spongy microspheres (NF-SMSs) can mimic the fibrous structure of the ECM at the nanoscale, recapitulate cell–cell and cell-matrix interactions, and regenerate pulp-like tissues in situ [26,27]. Electrospinning is another fairly straightforward nanotechnology-based technique used in pulp regeneration. The surface morphology and curvature of electrospun fibres are beneficial to the adhesion of DPSCs, and the supermolecular structure of collagen bundles nucleated on electrospun fibres contributes to directing the fate of DPSC towards odontogenic lineages without other cytokine factors [92].

The development of multipurpose composite polymer scaffolds, such as polyhydroxybutyrate (PHB)/chitosan/bioglass nanofibers [93] and β-glycerophosphate (β-GP)-loaded PCL/polyethylene oxide (PEO) blend nanofibers [94], promotes a paradigm shift of scaffolds from biocompatible cell transporters or simple delivery vehicles to biofunctional and guiding matrices. Further work should focus on assessing the synergetic effects between different polymer compositions and the added active therapeutics and cells.

### 4.3. Growth Factors

Although stem cells can promote pulp regeneration based on their multi-potent properties, this regenerative process can be accelerated by the delivery of robust growth factors. Growth factors play a critical role in promoting the mobilisation and engraftment of exogenously transplanted stem cells to recipient tissues [28,95,96], accelerating the recruitment of endogenous stem cells to the injured sites and then improving their regenerative function, thus contributing to the stimulation of pulp regeneration [97]. Studies have shown that multiple growth factors, alone or in combination, have the potential to induce many biological effects of DPSCs, among which migration, proliferation, and differentiation are essential for pulp regeneration [98]. Growth factors used for pulp regeneration should meet the following requirements: pulp/dentin regeneration, vascular regeneration, and neuronal regeneration. Below, we list the growth factors that have been found to meet these criteria (Table 6).
(1)Pulp/dentin regeneration: BMP [99,100], bFGF [101,102,103], stem cell factor (SCF) [104,105], G-CSF [28,95,96] and SDF-1α [106,107,108].(2)Vascular regeneration: VEGF [44,109,110], PDGF [111], bFGF [101,102] and SDF-1α [107,108].(3)Neuronal regeneration: nerve growth factor (NGF) and brain-derived neurotrophic factor (BDNF) [112].

It has been reported that loading scaffolds with bFGF could promote the proliferation and migration of DPSCs, the generation of pulp-like tissues and the deposition of mineralised tissues after subcutaneous transplantation; notably, a higher density of neovasculature was observed in newly formed tissues [102]. We assume that a key single signalling molecule can act as a catalyst to initiate more complex signalling cascades involved in pulp regeneration. Since there is a critical need for quick vascularisation in engineered tissue to enable the viability of transplanted cells, the combined application of multiple angiogenic growth factors might be an effective choice [113]. Based on the hypothesis that angiogenesis is a priority in pulp regeneration followed by regeneration of odontoblastic and neural tissue, Kim et al. proposed a two-step chemotactic cell homing approach: first, delivery of angiogenic and chemotactic cytokines, bFGF and VEGF, followed by the combined delivery of bFGF, VEGF, or PDGF with a basal set of NGF and BMP7 [15]. Reinnervation is another crucial requirement to maintain the long-term vitality of regenerated pulp tissues on the basis of revascularisation. It has been confirmed that PDGF-BB, NGF, and BDNF contribute to the regeneration of pulp-like tissues with good vasculature and innervation in vivo [112]. However, the minimum subset of growth factors or even a single robust growth factor, together with the optimal dosage, concentration, and delivery methods, warrant further evaluation to assess clinical applicability.

Recombinant human bone morphogenetic protein-2 (rhBMP-2), a well-known growth factor, was approved by the Food and Drug Administration (FDA) as a bone graft substitute in 2002. However, increasing side effects have been reported from BMP2 usage, causing safety concerns, including postoperative inflammation, ectopic bone formation, osteoclast-mediated bone resorption, and even tumorigenesis [114]. Novel approaches to identify and predict growth factor-target interactions in the complex regenerative process are indispensable for precision medicine.

## 5. Preclinical Exploration and Clinical Status of Pulp Regeneration

### 5.1. Animal Models and Research Patterns

Appropriate animal models are the key to verifying new ideas, concepts, and strategies from basic research to clinical translation. Currently, animal models for pulp regeneration are mainly divided into three patterns: ectopic, semiorthotopic, and orthotopic regeneration [117].
(1)Ectopic regenerative models: The dorsum subcutaneous space of immunocompromised rodent species could simulate the deficient blood supply of the dental pulp cavity. In this model, exogenous stem cells are seeded in HA/TCP scaffolds under the dorsal subcutaneous space of mice or rats to determine whether they possess the ability to form pulp/dentin in vivo [35,36]. In consideration of ready availability, easy operation, and low expenses, ectopic transplantation is supposed to be the first and basic modality for studying ectopic pulp regeneration.(2)Semiorthotopic regenerative models: The tooth slice/fragment scaffold can be chosen as an active carrier to deliver stem cells and/or growth factors. Since they were implanted together into an ectopic location in immunodeficient animals, regeneration occurred inside a real tooth. Therefore, these tooth slice/fragment models are considered semiorthotopic for pulp regeneration [41,42,118]. Such an approach is relatively simple and has the advantage of providing an orthotopic-like regenerative environment as well as minimising experimental variables. Unfortunately, there are obvious disadvantages: (a) the blood supply and operative procedures are significantly different from clinical conditions, and (b) the regenerated tissues are mainly produced and populated by mouse cells.(3)Orthotopic regenerative models: The pulp tissues of large nonprimate animals, such as ferrets [119,120], dogs [121,122], and swine [123,124], are more accessible and similar to humans. Thus, the regenerative performance following root canals in these animals can completely mimic clinical conditions [123]. Additionally, the single-rooted cuspid of ferrets and dogs, as well as single-/multirooted teeth of swine, provides a relatively larger space for model establishment and image-taking. Therefore, this model holds the highest value in various preclinical efficacy and safety tests.

Certain challenges and barriers must be investigated and solved in preclinical animal models before large-scale clinical trials can occur. Here, we list three preclinical research patterns in and corresponding animal models for pulp regeneration (Table 7). 

### 5.2. Safety Assessment of DPSC-Based Pulp Regeneration

Some clinical studies have suggested that autologous DPSC transplantation may be efficacious and safe for pulp regeneration in humans; however, certain limitations have to be overcome, such as the limited availability of autologous discarded teeth and the high cost of quality control tests of individual cell products before transplantation [122]. Furthermore, the biological activity of autologous DPSCs is reduced in elderly patients and patients with systemic diseases, including diabetes and osteopenia, compared with younger patients or healthy controls. Under this circumstance, allogeneic DPSCs might be a promising alternative for therapy, although there is an increased risk of immunorejection. Other concerns regarding both autologous and allogeneic DPSCs are perhaps the latent tumour-promoting effect, potential contamination, and pathogen transmission in the current culture conditions.

#### 5.2.1. Immunorejection and Systemic Inflammatory Response

In addition to the well-proven proliferative and multidifferentiation abilities, DPSCs share immunomodulatory characteristics similar to those of other MSCs. For example, it has been reported that DPSCs can inhibit proinflammatory M1 macrophage function via the TNF-α/IDO axis [125]. Inflammatory and reparative responses are intricately linked in dental pulp tissues; thus, harnessing the inflammatory environment of the pulp and optimising the immunomodulatory functions of DPSCs makes it possible to accelerate the regenerative process. However, questions regarding graft immunorejection should be addressed to support the clinical application of allogeneic DPSCs [126,127].

After transplanting allogeneic DPSCs with mismatched dog leukocyte antigen (DLA) in pulpectomised root canals of dogs, researchers confirmed that there were no qualitative or quantitative differences in regenerative pulp tissues between mismatched and matched groups. In addition, no evidence of toxicity, systemic inflammatory response or other adverse events appeared in the dual allogeneic transplantation of DPSCs with mismatched DLAs. Multiple and allogeneic DPSC transplantation seems to be a safe and promising strategy for total pulp regeneration [122]. Ideally, considering the large numbers of discarded teeth extracted from genetically diverse populations, adequate levels of clinical-grade human leukocyte antigen (HLA) isotype matching of DPSCs of patients may be achieved, and the establishment of an allogenic DPSC bank following good manufacturing practice (GMP) conditions should be feasible to facilitate therapeutic use [128].

#### 5.2.2. Tumorigenicity

Since DPSCs must be cultured, expanded, tested, and stored in vitro before therapeutic reintroduction into patients, it is critical to confirm whether prolonged manufacturing of DPSCs would trigger spontaneous mutations that may lead to uncontrolled growth or even aggressive tumours in recipients [129]. Wilson et al. reported that immortalised DPSC lines generated in their laboratory, whether spontaneously or induced, do not appear to have oncogenic potential or genomic instability, both in vitro and in vivo [129]. Consistently, Shen et al. [130] suggested that DPSCs were more resistant to oncogenic transformation compared with BMSCs. In addition, it is worth noting in this study that methylation and phosphatase and tensin homologue (PTEN) activation render DPSCs less susceptible to tumorigenesis, indicating that some specific epigenetic factors should be considered when applying DPSC-based therapies to pulp regeneration.

### 5.3. Optimised Strategies for Pulp Regeneration

#### 5.3.1. Novel Culture Methods for DPSCs

In contrast to conventional two-dimensional adherent culture systems, three-dimensional spheroid culture systems have become increasingly widely accepted as a better choice that contributes to higher proliferation, self-renewal capacity, and stemness maintenance of cells [55,69]. Novel NF-SMS, as both an injectable cell carrier and controlled growth factor delivery system, significantly enhanced the attachment, proliferation, and odontogenic differentiation of DPSCs and SCAPs, and achieved successful pulp regeneration in a full-length root canal with a large number of blood vessels in vivo [26,27,131,132]. Additionally, DPSC-laden gelatine methacryloyl (GelMA) microspheres could regenerate more vascularised pulp-like tissues after subcutaneous implantation in a nude mouse model compared with the DPSC-laden bulk GelMA group, and exhibited better degradability and cryopreservability [133].

The microenvironment, consisting of cell populations, ECM structures and soluble factors, exerts a significant influence on the physical behaviours, regulatory functions, and therapeutic effects of stem cells. Cell sheets and cell aggregates, retaining original native ECM and growth factors, are beneficial to the stable engraftment and long-term viability of transplanted DPSCs. Transplanted cell sheets and cell aggregates spontaneously release growth factors and extracellular vesicles, further contributing to pulp regeneration [32,40].

#### 5.3.2. Exosome-Mediated Pulp Regeneration

Recent studies have demonstrated that DPSCs affect tissue repair largely via replacement of damaged cells and their paracrine factors to stimulate host cells. Exosomes, important paracrine mediators, contain a broad spectrum of bioactive molecules derived from parental cells, including mRNAs, miRNAs and proteins. They can transfer important molecules to target cells and modulate cell functions [134,135]. Notably, DPSC-derived exosomes (DPSC-exos) can mimic the therapeutic benefits of parent cells while avoiding overt immune reactions and loss of stemness during expansion, storage, and transportation in vitro.

Exosomes derived from osteogenic DPSCs (DPSC-OD-exos) promoted the odontogenic differentiation of DPSCs via transfer of miR-5100, miR-27a-5p, miR-652-3p, and other miRNAs. DSPC-OD-exos also triggered the regeneration of pulp-like tissue in a semiorthotopic model, which seemed to be a safer alternative approach to DPSC-based pulp regeneration [33,136]. DPSC-exos also have the potential to induce endothelial cell proliferation and pro-angiogenic factor expression, playing an important role in dentin formation and angiogenesis [137]. It has been reported that cell type-specific exosomes can induce lineage-specific differentiation of stem cells [138]; thus, the attachment of dental stem cell-derived exosomes, including DPSC-exos, SHED-exos, and HERS-exos, to biomaterials such as collagen gel may be a more efficient and secure alternative option for pulp regeneration [139,140], while the exact molecular mechanisms remain to be investigated [141].

#### 5.3.3. Cell Homing-Based Pulp Regeneration

Different from cell transplantation, the cell homing strategy creates a regenerative capacity depending on active signalling molecules, which are embedded in cell-free scaffolds and released continuously to recruit endogenous stem cells [142]. There are many critical molecules participating in cell homing, including SDF-1α, bFGF, VEGF, PDGF, SCF, and G-CSF [98]. Unlike the formation of periodontal-like tissues by chemotaxis-induced pulp revascularisation, the regeneration of pulp/dentin-like tissues via cell homing has been confirmed in vivo. PDGF-BB, recognised as a potent chemoattractant, could promote the regeneration of highly vascularised pulp-like tissues surrounded by dentin-like mineralised tissues after subcutaneous injection into mice [111]. It was proposed that Alx3 could promote parenchymal and stromal regeneration of tooth tissue by direct transactivation of the Wnt3a and VEGF promoters; thus, the Alx3/Wnt3a axis might represent a pivotal signalling pathway in the regeneration of pulpodentinal complex. Wnt3a was shown to direct and regulate the survival, migration, proliferation, and rapid differentiation of DPSCs [143,144,145]. After infusing recombinant human Wnt3a in collagen gel into clinically treated root canals of miniswine, the regeneration of vascularised pulp-like tissues and mineralised dentin-like structures was observed, probably through the recruitment of endogenous cells and promotion of cell survival in vivo [34]. Researchers transplanted Alx3-restored DPSCs into tooth fragments under the dorsum of mice and achieved similar newly formed pulp/dentin-like tissue. The above results demonstrated that cell homing might serve as a contrasting regenerative approach while sharing common pathways and similar outcomes with cell transplantation.

However, the delivery of growth factors in endogenous cell homing is not as complex and costly as exogenous cell transplantation, which is more clinically translatable from a technical and economical point of view in light of the avoidance of cell preparation before application, as well as decreasing the risks of immunorejection and tumorigenicity after application [98,101]. By exploring potent combinations of growth factors and advances in stem cell recruitment, cell homing provides a better alternative approach and potentiates the feasibility and efficiency of pulp regeneration. The specific strengths and weaknesses of the three optimised strategies are shown in Figure 4.

### 5.4. Clinical Trials

Through the improvement of the biological understanding of dental pulp, abundant basic research, and preclinical explorations in the last decade, the therapeutic promise of DPSCs in preclinical studies has been translated to a clinical setting. Nakashima et al. [146] transplanted autologous DPSCs combined with GMP-grade G-CSF into five adult patients with irreversible pulpitis, among whom four patients demonstrated a robust positive response on electric pulp testing after four weeks. Eventually, cone beam computed tomography (CBCT) demonstrated functional dentin formation in three of the five patients, while clinical and laboratory evaluations demonstrated no adverse events or toxicity.

Xuan et al. [31] implanted autologous canine SHED aggregates into necrotic immature permanent incisors of thirty paediatric patients, achieving functional whole dental pulp regeneration with vasculature and innervation, both in clinical examination and histological analysis, as well as an odontoblast layer lining. At a 24-month follow-up examination, no complications, such as transplantation rejection and inflammation, had occurred. The success of the first full-length dental pulp regeneration has subverted traditional “standardized therapy” of RCT for pulpal and periapical diseases, which is of great significance in dental stem cell-based clinical translation [142].

Meza et al. [147] later on extracted inflamed dental pulp from a mature permanent tooth diagnosed with irreversible pulpitis, then isolated and cultured DPSCs in a GMP-grade laboratory. Expanded DPSCs along with leukocyte platelet-rich fibrin (L-PRF) obtained from the patient’s blood were introduced into the instrumented and disinfected root canal after a month. The 6-month and 3-year follow-ups revealed that the treated tooth was responsive to the cold and electric pulp test. The vitality test performed indicated low blood perfusion units. Although efficient and safe, personalised cell therapy using autologous DPSCs or SHEDs might be confronted with restricted translation to daily clinical practice because of the high cost associated with autologous therapy and the need of a GMP laboratory.

Interestingly, Feitosa et al. [148] developed a novel method of endodontic therapy. They performed dental pulp autotransplantation in three single-rooted premolars needed for RCT, from patients’ own third molar. At the 3- and 6-month follow-ups, positive pulp vitality and regression of periapical lesions were verified. After 9–12 months, all teeth were revascularised as determined by Doppler imaging. Dental pulp autotransplantation seems an innovative and feasible procedure for clinical application, which still needs major clinical trials.

## 6. Future Prospects of Pulp Regeneration

Recently, related studies on pulp regeneration have made significant progress, and some laboratory experiments have been translated into clinical therapeutic applications. For example, cell-based transplantation has been evaluated in pilot clinical trials, and cell homing assessments are ongoing in preclinical animal models. However, much effort should be focused on precision medicine, and the following translational challenges will need to be addressed for pulp regeneration in the future.

### 6.1. Challenge #1: The Reconstruction of Precisely Layered and Highly Ordered Dental Pulp Structures

Transplantation of a multilayer scaffold composite, namely, a well-aligned combination of competent stem cells, cytokines, and biomaterials, may facilitate the formation of well-organised and functional pulp tissue [124]. The recruitment of endogenous cells and signalling molecules via cell homing might also be a simple and effective approach for the regeneration of these complex tissues [15,34].

### 6.2. Challenge #2: Specific Therapeutic Procedures with Different Indications

Residual microorganisms in root canal systems may lead to unsuccessful treatment; thus, safer and more efficient disinfection strategies, such as nanobubble-enhanced antimicrobial agents, are needed in patients with severe periapical lesions rather than routine disinfection [149].

### 6.3. Challenge #3: Personalised Cellular Therapy for an Elderly Individual or an Individual with Severe Systemic Disease

Clinical pulp regeneration in mature permanent teeth has not been achieved due to the narrow apical foramen, insufficient blood supply, reduction in healing ability, and limited availability of pulp tissue with age. It is necessary to look for alternative cell sources, such as BMSCs, ADSCs, and other efficient subpopulations, combined with appropriate signalling molecules, thereby enabling pulp regeneration under low nutrition conditions and in cases of poor healing ability. Meanwhile, the establishment and extension of allogeneic DPSC banking [128], as well as some specific protocols such as trypsin pretreatment [20], would confer benefit to the optimisation of “precise medicine” and “personalized medicine” in the endodontic field.

## 7. Summary

Based on extensive preclinical studies and limited clinical trials, the functional regeneration of a pulpodentinal complex has been proved feasible and conceivable [31,111,146,147,148]. Importantly, the many years of basic research in this field have set the foundations for a now-exploding biotech and industrial activity devoted to turning such knowledge into treatments for pulpal and periapical diseases. Basic scientific research outcomes can ultimately inform clinical diagnosis and treatment. It is believed that pulp regeneration will eventually become a clinical reality and will have a far-reaching impact on the treatment of pulpal and periapical diseases, and even on the regeneration of whole teeth and other tissues in the future. The next few years will see whether its promise is fulfilled; exciting times lie ahead.

## Figures and Tables

**Figure 1 ijms-22-08991-f001:**
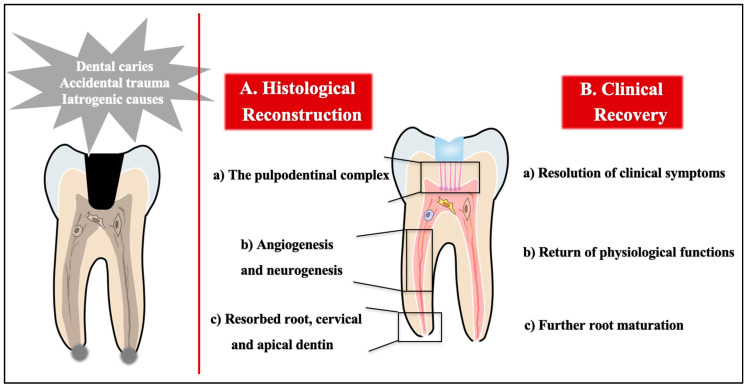
Histological and clinical goals of pulp regeneration. Dental pulp may be damaged by dental caries, accidental trauma, or iatrogenic causes. The success of pulp regeneration should be defined by (**A**) histological and (**B**) clinical measures. Reconstruction of the pulpodentinal complex through angiogenesis and neurogenesis could yield histologically successful regeneration following cervical and apical dentin formation. Clinical success includes elimination of clinical symptoms such as pain and bone resorption, return of physiological functions, and increased root wall thickness and/or increased root length.

**Figure 2 ijms-22-08991-f002:**
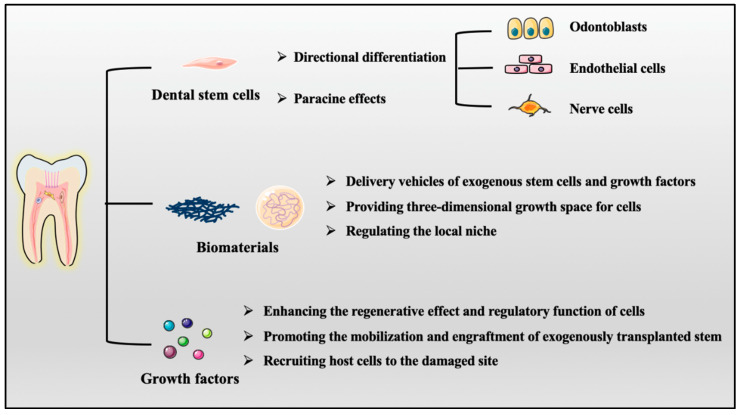
Three key elements in pulp regeneration: dental stem cells, biomaterials, and growth factors.

**Figure 3 ijms-22-08991-f003:**
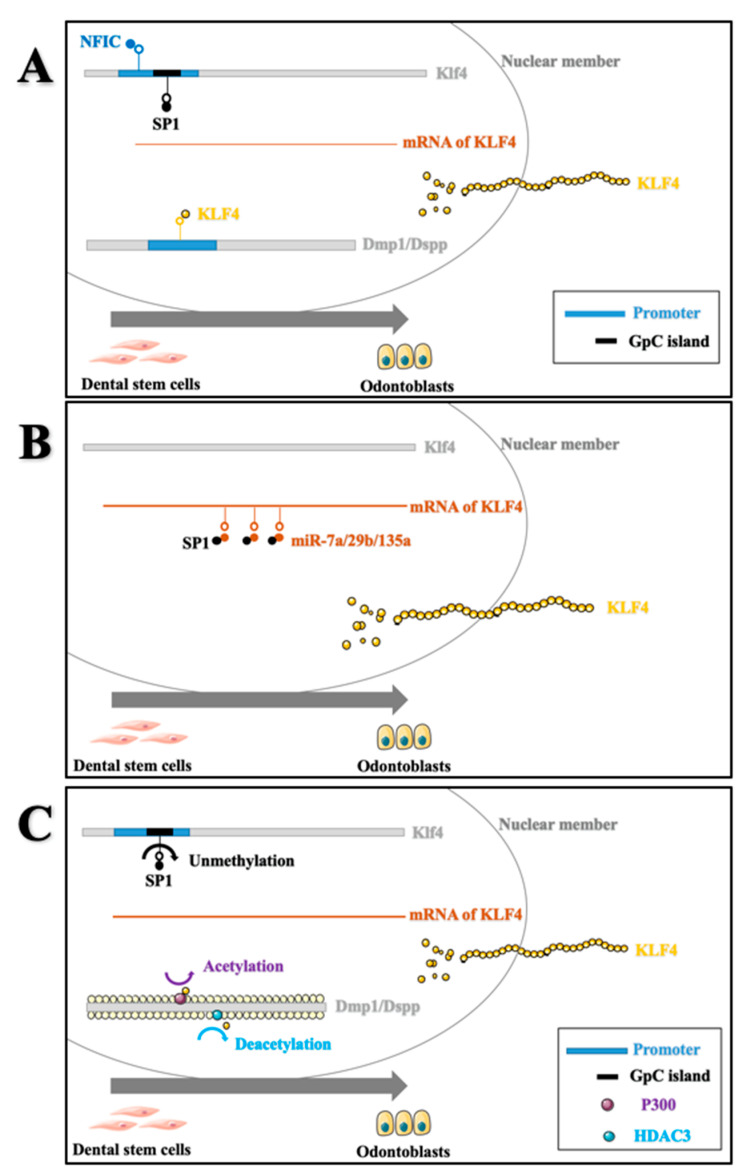
Transcriptional (**A**), posttranscriptional (**B**), and epigenetic (**C**) regulation of KLF4 in DPSCs. KLF4 could directly upregulate the expression levels of Dmp1 and Dspp by binding to their promoters; similarly, NFIC and SP1 could regulate KLF4 expression by binding to the promoter region of Klf4. Sp1 was able to function as a ceRNA of Klf4 as well. Demethylation of the SP1/KLF4 binding motif enhanced the efficiency of SP1 binding and transcriptional regulation of KLF4. Additionally, KLF4 could recruit the histone acetylases P300 and HDAC3 to transactivate the expression of Dmp1 and Sp7. Complicated regulatory networks among the Sp1/Klf4/Dmp1, Sp1/Klf4/Dspp, and Sp1/Klf4/Sp7 axes need further investigation.

**Figure 4 ijms-22-08991-f004:**
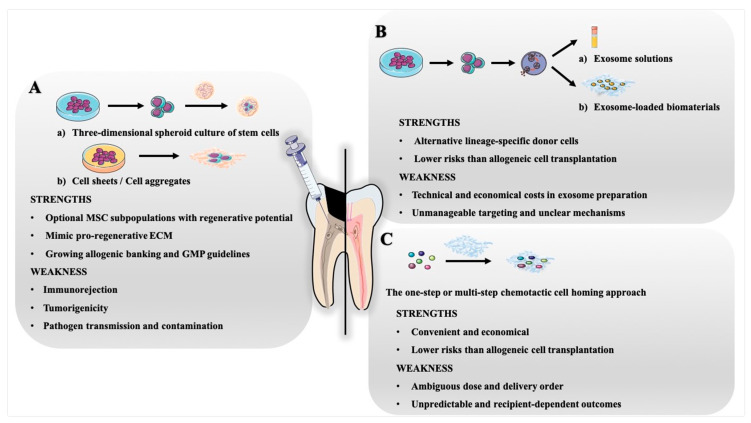
Comparison of three optimised strategies for pulp regeneration. (**A**) Novel culture methods for transplanted dental stem cells, such as three-dimensional spheroid culture systems, cell sheets, and cell aggregates, are beneficial for determining the optimal regenerative subpopulations and mimic pro-regenerative ECM, while some limitations of allogeneic transplantation cannot be avoided. (**B**) Exosomes used for pulp regeneration could be selected from alternative lineage-specific donor cells; however, high costs in isolation, identification, and administration before application may limit their large-scale production. Further investigations must focus on precise targeting and uptake testing. (**C**) An exogenous growth factor-mediated cell homing approach would be an operable and economical choice for pulp regeneration, although a wide variation in growth factor types, applied dosages, and delivery orders among recipients may lead to unpredictable outcomes.

**Table 1 ijms-22-08991-t001:** Comparison of RCT, pulp revascularisation and pulp regeneration via treatment procedures, root canal fillings, outcomes, complications, and limitations.

Approaches	Root Canal Therapy (RCT)	Pulp Revascularisation	Pulp Regeneration
Treatment procedures	Root canal preparation, disinfection and filling [4]	Sufficient chemical disinfection, induction of bleeding into the canal and careful sealing [9]	Based on tissue engineering strategies, such as exogenous cell transplantation and endogenous cell homing [7,8]
Root canal fillings	Inert materials such as gutta-percha	Periodontal-like tissue [11]	Pulp/dentin-like tissue
Outcomes	Nonvital teeth	Uncertainty (new dentin formation and continued root development)	Vital teeth (restored homeostasis and natural defence that promote tooth survival)
Complications and limitations	Technology sensitivity [17] Reinfection [18] Loss of pulp function [19] Age limitations [20]	Calcification and discolouration [21] Individual differences and unpredictable outcomes [22] Lack of standardised clinical evaluation protocols [22] Unclear effective molecules in blood clots	Strict case selection and unpredictable outcomes Immunorejection, potential contamination, pathogen transmission and tumorigenesis Storage methods and application expenses

**Table 2 ijms-22-08991-t002:** Therapeutic applications of the three essential elements.

Involved Strategies	Exogenous Stem Cells	Biomaterials	Growth Factors	Examples	Situations
Cell transplantation	Yes	Yes	Yes	Pulp CD105+ cells with SDF-1 and collagen scaffold [13]	Inferior vitality and limited sources of donor cells (elderly individuals, people with systemic diseases)
	Yes	Yes	None	DPSCs within nanofibrous spongy microspheres [26,27]	
	Yes	None	Yes	DPSCs mobilised by G-CSF [28]	
	Yes	None	None	DPSCs electrotransfected with Gdf11 [29], iPS cells [30], SHED aggregates [31], and DPSC injection solution [32]	
Stem cell-derived extracellular vesicles	None	Yes	Yes	DPSC-derived exosomes within collagen sponges [33]	Avoiding the dosage, delivery and safety concerns of exogenous stem cells and growth factors
Cell homing	None	None	None	Pulp revascularisation [9]	Immature permanent teeth
	None	None	Yes	Autologous platelet concentrates applied in pulp revascularisation [25]	Immature and mature permanent teeth
	None	Yes	Yes	A combination of Wnt3a, BMP7 and collagen gel [34]	Inferior vitality and limited sources of donor cells (elderly individuals, people with systemic diseases)

CD105: Endoglin; SDF-1: Stromal cell-derived factor-1; G-CSF: Granulocyte-colony stimulating factor; Gdf11: Growth/differentiation factor 11; iPS cells: Induced pluripotent stem cells; SHEDs: Stem cells from human exfoliated deciduous teeth; BMP7: Bone morphogenetic protein 7.

**Table 3 ijms-22-08991-t003:** DPSC- and SHED-mediated pulp regeneration from the bench to the clinic.

Timeline	Clinical Significance	Methods	Species/Implantation Sites
2000 [35]	Discovery of DPSCs	DPSC transplantation with HA/TCP powder	Mice/subcutaneous space
2003 [36]	Discovery of SHEDs	SHED transplantation with HA/TCP powder	Mice/subcutaneous space
2004 [29]	The first DPSC transplantation in a dog model	Gdf11-transfected DPSCs cultured as a pellet	Dogs/root canals after partial pulpectomy
2007 [41]	Successful revascularisation after DPSC transplantation	DPSCs seeded in human tooth slices	Mice/subcutaneous space
2010 [42]	The goal of functional pulp/dentin regeneration and major concerns, such as less organised dentinal tubules and vascularity, are addressed	DPSC transplantation with PLG scaffolds and human tooth root fragments	Mice/subcutaneous space
2010 [15]	Cell homing approach for pulp regeneration	Endodontic treatment of human teeth with a cytokine-adsorbed collagen gel, such as that containing bFGF, VEGF, or PDGF	Mice/subcutaneous space
2011 [13]	The first complete orthotopic pulp regeneration replete with angiogenesis and neurogenesis	Pulp CD105+ cells/SDF-1/collagen scaffold	Dogs/root canals after pulpectomy
2018 [31]	The first randomised clinical trial using autologous SHEDs for pulp regeneration	Autologous SHED aggregates containing cells and ECM	Immature permanently injured incisors
2020 [32]	Registration of DPSC injection solution for clinical applications	DPSC injection solution	Patients

HA/TCP: Hydroxyapatite/tricalcium phosphate; PLG: Poly-d,l-lactide/glycolide; bFGF: Basic fibroblast growth factor; VEGF: Vascular endothelial growth factor.

**Table 4 ijms-22-08991-t004:** Characteristics of stem cells and their application in pulp regeneration.

Stem Cells	Advantages	Disadvantages	References
DPSCs	Tissue specificity Potent capacity for vascular formation Easy accessibility	Available pulp tissue is reduced under pathological conditions	[40,43,44,45]
SHEDs	Tissue specificity Higher proliferative activity Lower immunogenicity Osteoinductive capacity Higher neurogenic differentiation capacity	Deficient in the specific formation of the extracellular matrix (ECM)	[46,47,48]
PDLSCs	Potent capacity for differentiation into cementoblasts and osteoblasts Prone to form directionally arranged fibres	Low cell numbers in original cultures Low neurogenic differentiation capacity	[37,49]
DFPCs	Participate in the regeneration of cementum and periodontal ligament	Low cell numbers in original cultures Isolation difficulty Prone to differentiate into specific periodontal lineages	[50,51]
SCAPs	Strong osteogenic and angiogenic differentiation potential Anti-inflammation function Dynamic secretome Prone to differentiate into primary odontoblasts rather than reparative dentin	Low cell numbers in original cultures Difficult to isolate and distinguish from PDLSCs, DFPCs and HERS	[52,53]
HERS	Participate in tooth root and periodontium formation	Low cell numbers in original cultures A lack of appropriate culture methods	[54,55]
BMSCs	Participate in the formation of osteoid tissue and pericyte-supported capillaries	Low neurogenic differentiation capacity Invasive surgical procedures required to obtain cells	[43,44,45,56]
ADSCs	Abundance Ease of harvest and processing	Weak odontogenic and angiogenic differentiation potential	[43,44,45]

**Table 5 ijms-22-08991-t005:** Specific surface markers of DPSCs and their functions.

Surface Markers	Authors and Year	Events and Results	Specific Functions
Flk-1+ (VEGF-R2+)	Aquino et al. (2007) [59]	Giving rise to the regeneration of vascularised tissues by synergistically differentiating into osteoblasts and endothelial cells.	Angiogenesis
CD31−/CD146−	Iohara et al. (2009) [58,60]	Total pulp regeneration with capillaries, nerve cells, and expression of pro-angiogenic factors.	Dentinogenesis, angiogenesis and neurogenesis
CD105+	Iohara et al. (2009) [13,61]	Larger regenerated pulp tissues including nerves, vasculature and dentin formation.	Potent trophic effects on neovascularization
STRO-1+	Yu et al. (2010) [62]	Spontaneous differentiation into odontoblasts, osteoblasts, and chondrocytes.	Dentinogenesis, osteogenesis and chondrogenesis
NG2+	Zhao et al. (2014) [63]	Participation in emergency responses such as injury repair of pulp tissues rather than physiological homeostasis.	Actively involved in reparative dentin formation
CD271+ (LNGFR+)	Alvarez et al. (2015) [64]	The isolation of a relatively large population of DPSCs (10.6%) with the strongest odontogenic and chondrogenic potential.	Dentinogenesis and chondrogenesis
CD271^Low+^CD90^High+^	Yasui et al. (2016) [65]	The most clonogenic population in dental pulp capable of adipogenic, osteogenic, and chondrogenic differentiation in vitro and promotion of new bone formation in vivo.	Long-term viability, clonogenicity and osteogenicity
CD90+ (Thy-1+)	An et al. (2018) [66]	Contributes 30% of the odontoblasts and pulp cells during early postnatal development, as well as when the tips of the incisors are clipped.	Corresponding to a rapid growth rate increase in both established and re-established tooth length
αSMA	Vidovic et al. (2017) [67]	A second generation of odontoblasts during reparative dentinogenesis, also a small contribution to odontoblasts during primary dentinogenesis.	Dentinogenesis
PD-1	Liu et al. (2018) [68]	Controlling cell proliferation and multipotential differentiation of DPSCs.	Stemness maintenance
CD24a+	Chen et al. (2020) [69]	Enhanced osteogenic/odontogenic differentiation capabilities, regenerative dentin and neurovascular-like structures formation in vivo.	High proliferative and self-renewal capacity, highly efficient regeneration of pulpodentinal complex-like tissues

CD73: 5′-ectonucleotidase; CD271 (LNGFR): Low-affinity nerve growth factor receptor; CD90 (Thy-1): Glycosylphosphatidylinositol-anchored glycoprotein; α-SMA: α-smooth muscle actin.

**Table 6 ijms-22-08991-t006:** Growth factors used for pulp/dentin regeneration.

Growth Factors	Authors/Years	Employed Cells	Mechanism	Test Model	Functions
BMP2, 4, 7 *	Nakashima et al. (2003) [99]	DPSCs	BMP/Smad pathway	In vitro	Pulpodentinal complex, periodontal and craniofacial regeneration
	Yang et al. (2012) [100]	DPSCs	VEGFA/VEGFR2 pathway	In vivo	Dentinogenesis and angiogenesis
bFGF	Suzuki et al. (2011) [101]	None	None	In vivo	Induction of recellularisation and revascularisation
	Yang et al. (2015) [102]	DPSCs	None	In vivo	Dentinogenesis, angiogenesis and neurogenesis
	Chang et al. (2017) [103]	DPSCs	MEK/ERK pathway	In vitro	Promotion of proliferation, differentiation, and matrix production of DPSCs
VEGF	Silvana et al. (2007) [41]	DPSCs	SDF-1α activation and angiogenic cascade initiation	In vivo	Dentinogenesis and angiogenesis
	Zhang et al. (2016) [109]	DPSCs	Wnt/β-catenin pathway	In vivo	Vasculogenesis and angiogenesis
	Bae et al. (2018) [110]	DPSCs	LOX activation	In vitro	Odontogenesis and angiogenesis
SCF	Pan et al. (2013) [104]; Ruangsawasdi et al. (2017) [105]	DPSCs	PI3K/Akt and MEK/ERK pathway	In vivo	Promotion of DPSC migration, neovascularisation, and collagen remodelling
G-CSF *	Nakashima et al. (2013) [28,95,96]	MDPSCs	G-CSF/G-CSFR	In situ	Inhibition of apoptosis, promotion of cell survival, suppression of inflammation, and induction of angiogenesis and neurogenesis
SDF-1α	Li et al. (2015) [106]	SCAPs	SDF-1α/CXCR4 axis	In vitro	Chemotactic function
	Yang et al. (2015)107	DPSCs	Autophagy activation	In situ	Mineralisation, neovascularisation and chemotactic function; probably innervation
	Nam et al. (2017) [108]	DPSCs	SDF-1α/CXCR4 axis	In vivo	Angiogenesis
PDGF-BB *	Zhang et al. (2017) [111]	DPSCs	PI3K/Akt pathway	In vivo	Enhancing hDPSC proliferation, migration, angiogenesis, and odontogenic differentiation
L-WNT3A	Zhao et al. (2018) [115]; Chen et al. (2020) [116]	DPSCs and odontoblasts	Wnt/β-catenin pathway	In vivo	Prosurvival and antiapoptic effects, as well as induction of more tertiary dentin formation

MEK: Mitogen-activated protein kinase; ERK: Extracellular-signal regulated kinase; LOX: Lysyl oxidase; PI3K: Phosphatidylinositol 3 kinase; MDPSCs: Mobilised dental pulp stem cells; *: FDA approval.

**Table 7 ijms-22-08991-t007:** Three preclinical research patterns and commonly used animal models in pulp regeneration.

Research Patterns	Animals	Scaffold /Implantation Sites	Advantages	Limitations	Application Conditions
Ectopic model	Mice/rats [35,36]	HA/TCP; the dorsum subcutaneous space	Small, convenient, and easily maintained Readily available Inexpensive No concern of rejection because of the immunocompromised condition	Unconvincing results Only used in simple experiments	The first and basic investigation before an intensive study
Semiorthotopic model	Mice/rats [41,42,118]	Tooth slices/fragments; the dorsum subcutaneous space	Relatively simple procedures while providing an orthotopic-like environment for pulp regeneration Singular and minimised experimental variables	Differences in blood supply and operative procedures between clinical conditions Regenerated tissues produced by both human and mouse cells	Mechanistic and translational studies with DPSCs, such as those investigating signalling regulation and optimised transplantation conditions for pulp regeneration
Orthotopic model	Ferrets [117,119,120]	Various biomaterials; root canals after pulpotomy or pulpectomy	Accessibility and large size of the single-rooted cuspid Fewer ethical objections, less expensive and more readily available than other large animals	Domesticated pets in some cultures	Other large animals are unavailable or have limited feeding conditions
	Dogs [121,122]		Relatively large teeth, easily accessible and radiographed Easier to explore exposed time and depth of root canal enlargement	Dog DPSCs have differences in regenerative potential and immunogenicity with hDPSCs Difficulty of handling High cost to purchase and feed Ethical problems	Complete pulp regeneration with periapical disease Preclinical efficacy and safety studies in special populations
	Swine [123,124]		Including single-/multirooted teeth Swine DPSCs and microenvironments are similar to human counterparts	Complicated and challenging root canal morphology Difficulty of tooth sample processing and decalcification Difficulty of handling High cost to purchase and feed Ethical problems	Heterogeneous population of DPSCs for pulp regeneration Natural dentin structure with organised dentinal tubules Bioroot and apical dentin regeneration

## Data Availability

All data are included in this published article.

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
