# Peer review of "Functional Dental Pulp Regeneration: Basic Research and Clinical Translation"

_ijms, 2021, doi:10.3390/ijms22168991_

Round 1

Reviewer 1 Report

The paper entitled “Functional Dental Pulp Regeneration: Basic Research and Clinical Translation” is a descriptive contribute that  gives a general overview on mediated tissue engineering pulp regeneration.

The work  provides original data derived from preclinical experiments based on various animal models and research strategies .

INTRODUCTION

Overall well structured, it provides all the information necessary to understand the scientific background, the knowledge gap and the objectives of the paper.

REVIEW

It is a simple descriptive review that is limited to a fairly scholastic description of the different possibilities of pulp regeneration, highlighting in particular the advantages and limitations of the different methodologies currently being tested.

Overall, the work can be considered interesting as it describes in a fairly clear way the new alternative therapeutic strategies in the endodontic field that could be applicable in the near future, however, from the point of view of scientific evidence there is no relevant data as considering the scarce consistency of the available contributions, no statistical evaluation was performed.

SUMMARY

This part of the work is far too general; should be rewritten providing some more detailed information on the different possible ways of pulp regeneration and above all on what may be the future possibilities of use in clinical practice

Author Response

1. INTRODUCTION

We greatly appreciate the reviewer’s efforts to carefully review the paper and the valuable comments offered.

2. REVIEW

We appreciate the reviewer’s expertise assessment and insightful evaluation. We provide an integrative and narrative review, and have a systematized and described methodology for selecting the bibliography. In the revised paper, we have added the search strategy and scope of the review, which now reads:

2. Literature search and scope of the review

An electronic search was performed on PubMed, Web of Science and Scopus Library databases of English language publications from inception to April 20, 2021. The related key words included "Dental pulp stem cells (DPSCs)", "Dental pulp regeneration", “Dental pulp repair”, "Tissue engineering", "Clinical translation", and “Regenerative endodontics”. However, due to the scope and extent of this search, a wide ranging comprehensive narrative review of functional dental pulp regeneration rather than a systematic review was undertaken. ” (please see line 88-95). We hope the reviewer might agree with this perspective.

3. SUMMARY

Thank you for your suggestions. Actually, we had clarified the current challenges and promising solutions in pulp regeneration in section 6. Future prospects of pulp regeneration. In the manuscript, we also revised the summary in section 7 as suggested and hope the reviewers might agree with this perspective (please see line 624-633).

Reviewer 2 Report

Commends to the authors for a well written and comprehensive review !

The text organization .tables and schems are clear and contribute to profund understanding of the topic by students, researchers and endodontists 

Author Response

We would like to express our deep thanks for taking the time to review our manuscript and for your positive comments.

Reviewer 3 Report

First, congratulate the authors. Very important work, well-structured and very comprehensive, in one of the most current themes of contemporary dentistry, which must be conservative, individualized and biologically and physiologically oriented.

However, I would like you to clarify some doubts:

  1. As a review work, you should indicate what type of review was performed, and which search bibliography strategy was used (sources, search period, keywords, inclusion and exclusion criteria, etc). Even if it was a simple, non-systematic, narrative review, it should have a systematized and described methodology for selecting the bibliography. This website can be very useful (https://cihr-irsc.gc.ca/e/41382.html).

  1. Still in this point. Although the bibliography is extensive and updated, it could still be complemented with some works in the field of animal research.

  1. Line 48 to 50: refer that revascularization treatments are already frequently used in the clinic! There is some published work that indicates what percentage of these treatments are carried out in the daily routine of dentistry? Because I believe they are only used in exceptional situations and by a restricted group of professionals (but that's just my sensibility).

  1. Line 53 and 54: Refer to “in descending order of importance”! Is it not the contrary? “ascending order of importance”?

  1. Line 90: “Technology sensitivity: it is difficult for interns to ….”? Who wish to refer as "interns"?

  1. Line 93: “Instrument separation” or “instrument fracture?

  1. Line 153: In the key elements in pulp regeneration (dental stem cells, biomaterials and growth factors) I think that one could and should add and emphasize the need for vascularization, otherwise nothing will work without this crucial nutritional contribution.

Author Response

1. Thank you very much for your insightful evaluation. Your suggestion is of great significance to our work. We have added the search strategy in the review (line 88-95). It reads as follows:

 2. Literature search and scope of the review

An electronic search was performed on PubMed, Web of Science and Scopus Library databases of English language publications from inception to April 20, 2021. The related key words included "Dental pulp stem cells (DPSCs)", "Dental pulp regeneration", “Dental pulp repair”, "Tissue engineering", "Clinical translation", and “Regenerative endodontics”. However, due to the scope and extent of this search, a wide ranging comprehensive narrative review of functional dental pulp regeneration rather than a systematic review was undertaken.”  

We have also searched website you recommended and updated references in “5.1 Animal models and research patterns”. Please see reference 119-124.

2. Thanks. There are no relevant reports on the percentage of revascularization treatments in dental clinic. It may be our inappropriate expressions that caused the misunderstandings. We have changed “Formally approved and commonly used in the clinic” to “Formally approved and used in the clinic” and hope the reviewers might agree with this perspective (line 53-54).

3. Thanks for your question. According to AAE 2016, the degree of success of Regenerative Endodontic Procedures is largely measured by the extent to which it is possible to attain primary, secondary, and tertiary goals:

(1) Primary goal: The elimination of symptoms and the evidence of bony healing.

(2) Secondary goal: Increased root wall thickness and/or increased root length. (desirable, but perhaps not essential)

(3) Tertiary goal: Positive response to vitality testing. (which if achieved, could indicate a more organized vital pulp tissue)

In the revised manuscript, we have modified this part to make it clear and hope the reviewers might agree with this perspective (please see line 61-65).

4. We appreciate your comment. We have changed “interns” to “general dentists” to make it more accurate. Please see in line 107.

5. Thank you for your suggestion. We have changed “instrument separation” to “instrument fracture” in the revised manuscript. Please see in line 109.

6. Thank you very much for your valuable comment. As suggested, we have changed “A suitable combination of these three elements enables the manipulation of a biomimetic microenvironment, which is necessary for clinical application in pulp regeneration” to “A suitable combination of these three elements enables the manipulation of a biomimetic microenvironment containing a well-functioning vascular system, which ensures nutrient supply, waste removal, inflammatory response, and subsequent regeneration for the pulp”. Please see lines 166-168 of the revised manuscript.

Reviewer 4 Report

Line 16.  The use of the Christian cross is often interpreted as meaning a person has died. It is a good symbol to use in academic writing. 

Significant grammatical issues are present.  The manuscript is almost unreadable.

Formal definitions of pulp revascularization vs pulp regeneration needs to be stated in early in the introduction

Interns?? Did you mean to use another word?

Line 48.  Sentence is written poorly.  RCT teeth preserve function. I assume you mean extraction. Most English speakers use Function as  - it is of use.  

The paragraph starting at line 90. The numbering is confusing as the reader will assume these are references, not points.  The numbering does not add to the argument and can be removed

Line 95.  It is evident you misunderstand the use of function in dentistry.

Table 1. Information within table does not match the headings.  Also, inadequate use of referencing. 

Line 104.  It is too late to start discussing the definition after you have already pointed out its pitfalls.

Line 117.  Incorrect use of the term apexification. MTA does not form an apex. It used to create an apical barrier.

Line 118.  /0? Do you mean 0%?  But also, why is there a / ? Are you dividing values.  It doesn’t make sense.

Table 3 should not have alphabet dot points for one column but no alphabet dot points for the disadvantage column.  There is no value is having alphabet dot points in this context.

Table 6 should not have “/“ as no data.  Just leave it blank or write something like “No information is available”

Table 7 should be corrected as it is difficult to read the columns.  Again, alphabet dot points are unnecessary

Line 457 should have the citation rather than just the author.  This error seems repeated.  If you name the author, you still need to add the numeric citation.

References: Inconsistent abbreviation of journal titles.  Some journal titles are abbreviated and some are not.

References: Inconsistent capitalisation of titles.  Some are in title case and some are in sentence case.

Why are case reports being quoted?  These are the lowest forms of evidence.

-Altaii, M, Richards, L, Rossi-Fedele, G. Histological assessment of regenerative endodontic treatment in animal studies with different scaffolds: A systematic review. Dent Traumatol. 2017; 33: 235– 244. https://doi.org/10.1111/edt.12338

-Kahler B, Rossi-Fedele G. A Review of Tooth Discoloration after Regenerative Endodontic Therapy. J Endod. 2016 Apr;42(4):563-9. doi: 10.1016/j.joen.2015.12.022. Epub 2016 Feb 4. PMID: 26852148.

The latter is of clinical importance as patient focused outcomes.  Furthermore, the review below highlights the clinical indications. The current manuscript does not discuss calcium hydroxide apexification which, by some, is still viewed as a valid treatment. I disagree and the review below supports this.  If your manuscript does not explain how pulp revas or pulp regeneration could be valid, the reader will not care.

Duggal M, Tong HJ, Al-Ansary M, Twati W, Day PF, Nazzal H. Interventions for the endodontic management of non-vital traumatised immature permanent anterior teeth in children and adolescents: a systematic review of the evidence and guidelines of the European Academy of Paediatric Dentistry. Eur Arch Paediatr Dent. 2017 Jun;18(3):139-151.

Also, a lot of what is discussed is similar to 

Lin LM, Huang GT, Sigurdsson A, Kahler B. Clinical cell-based versus cell-free regenerative endodontics: clarification of concept and term. Int Endod J. 2021 Jun;54(6):887-901.

The study also seems to be missing

Dental Pulp Autotransplantation: A New Modality of Endodontic Regenerative Therapy—Follow-Up of 3 Clinical Cases

Author links open overlay panel

Victor PinheiroFeitosa

Author Response

Thanks to reviewers for the careful review and constructive comments. We have tried to address each of the concerns raised in the review, on a point by point basis, and incorporated these changes into the revised manuscript. In doing so, the paper has been improved and strengthened. Our responses are present in attachment.

Round 2

Reviewer 4 Report

The grammar is quite poor with many poorly written sentences. Please consider using Grammarly to correct the English. Otherwise, please find a professional proofreader to correct the English.

The referencing style is inconsistent. The page numbers. e.g. 377-390 vs 125-32

Table 6 should be author/year

Table 7 does not need vertical lines. 

Please check if the abbreviations should be in alphabetical order